# Barriers and Expectations of Adolescents Regarding the Identification and Management of Their Psychoactive Substance Use by Their General Practitioner

**DOI:** 10.3390/ijerph192013231

**Published:** 2022-10-14

**Authors:** Frédéric Fortin, Stéphanie Roche, Julie Dupouy, Pierre Bernard, Jonathan Lachal, Céline Lambert, Catherine Laporte

**Affiliations:** 1Département de Médecine Générale, UFR de Médecine et des Professions Paramédicales, Université Clermont Auvergne, F-63000 Clermont-Ferrand, France; 2Institut Pascal, CNRS, CHU Clermont-Ferrand, Clermont Auvergne INP, Université Clermont Auvergne, F-63000 Clermont-Ferrand, France; 3Département Universitaire de Médecine Générale de Toulouse, Université Paul Sabatier Toulouse III, 31062 Toulouse, France; 4UMR 1295 Inserm, Université Toulouse III, F-31000 Toulouse, France; 5MSPU de Pins Justaret, 31860 Pins Justaret, France; 6Service de Psychiatrie de l’Enfant et de l’Adolescent, CHU Clermont-Ferrand, F-63000 Clermont-Ferrand, France; 7UFR de Médecine et des Professions Paramédicales, Université Clermont Auvergne, F-63000 Clermont-Ferrand, France; 8Team DevPsy, CESP, Inserm, UVSQ, Université Paris-Saclay, F-94807 Villejuif, France; 9Unité de Biostatistiques, DRCI, CHU Clermont-Ferrand, F-63000 Clermont-Ferrand, France

**Keywords:** addiction, adolescent health, addictive behavior, general practice, physician–patient relations, primary care, substance-related disorders

## Abstract

Aims. General practitioners (GPs), who are the most frequently consulted health professionals by adolescents, play a key role in screening for psychoactive substance (PAS) use. The purpose of our study was to determine the barriers and expectations of adolescents regarding the identification and management of their PAS use by their general practitioner. Methods. Descriptive, cross-sectional study of a population of adolescents aged 12 to 17 years, followed up in general practice in France. Adolescents were recruited from general practice offices by open-access questionnaires. An opaque box was provided to ensure the anonymity of the adolescents. Results. A total of 277 adolescents were included: 155 girls, mean age 14.5 ± 1.7 years, 113 adolescents (41%) had used a PAS at least once in the past 12 months. Alcohol was the most used PAS, followed by tobacco and cannabis. Three groups were identified: the nonusers group (n = 134); the group of moderate users (n = 71); the group of users at risk of substance abuse or misusing (n = 38). Regardless of group, adolescents felt that their GP was attentive, responsive, competent, understanding, and took the time to ask the appropriate questions in their role. The at-risk group was less confident and less comfortable, and they felt more judged and more afraid of the GP telling their parents. Despite this, the at-risk group was the most willing to talk to their GP about their PAS. Almost half of the adolescents surveyed found it useful to use a questionnaire to discuss PAS. Conclusions. Reminding each consultation of the principles of the relationship of trust and confidentiality while maintaining an empathetic attitude would make it easier for GPs to remove adolescents’ inhibitions about communicating about their PAS use.

## 1. Introduction

In France, in 2017, 86% of 17-year-olds had experimented with alcohol, 59% with tobacco, 39% with cannabis, and 7% with an illicit drug other than cannabis [1]. Regular polydrug use (at least 10 uses in a month) concerned 9% of 17-year-olds [2], and almost 2% of young people are polyconsumers of tobacco, cannabis, and alcohol. On average, adolescents report their first tobacco cigarette use at 14.4 years of age [1], cannabis use at 15.3 years of age [1], and alcohol use at 13.4 years of age [3]. According to the 2015 ESPAC survey, regarding the situation of French adolescents compared to their European counterparts, their level of recent (in the last 30 days) tobacco use is higher than average: 26% vs. 22% (11th place out of 35 countries). The recent alcohol consumption of 16-year-olds in France is in line with the European average: 47% (15th place). Finally, at age 16, the French lead the European ranking for recent cannabis use (17%) [4]. Thus, the initiation of legal and illegal psychoactive substance (PAS) use occurs during adolescence [5]. The earlier in life the use of PAS begins, the higher the risk of abuse and/or development of dependence, especially if the use is repeated [6].

Given a specific vulnerability to the occurrence of misuse or its consequences, adolescents should be particularly monitored with respect to their level of PAS use [7,8]. Addiction is often initiated during adolescence when the still-incomplete decision-making control circuits have difficulty regulating emotional, impulsive, and conditioned responses [9,10]. PAS misuse during this period can cause specific damage to the brain maturation processes and neuropsychological development of the adolescent [8,10]. For all these reasons, adolescence is a period of particular vulnerability to the use of PAS, which requires regular vigilance and monitoring of their use and rapid intervention in case of misuse.

The consequences of this PAS use can be multiple, acute, or chronic. Alcohol use is associated with an increased risk of intentional and unintentional injuries, violence, mental health problems, and unprotected sex [6]. Illicit drug use is the second most important risk factor for youth health in high-income countries, measured in disability-adjusted life years [11]. Behaviors adopted during adolescence thus have important consequences for health in adulthood. Tobacco, alcohol, and illicit drug use contribute to 17% of the global burden of disease across all age groups [11].

In France, in 2009, during the last seven days of their professional practice, 69% of general practitioners (GP) saw in consultation at least one patient in the context of smoking cessation and 52% in the context of alcohol cessation. Almost two-thirds of GPs have been consulted by an opioid user during the year, and 59% have been consulted by a cannabis user [12]. However, the proportion of French GPs involved in treating people with substance use disorder remains very uneven and involves the following in decreasing order: tobacco, alcohol, cannabis, and opioids [13].

Moreover, GPs are the health professionals most frequently consulted by adolescents. Epidemiological data and experience show that the encounter between a GP and an adolescent is frequent. Although adolescent consultations are low (2.1 times per year on average for boys and 2.5 for girls), 75% of adolescents have seen a GP during the year [14]. The use of PAS by adolescents is therefore a concern for GPs because it is possible to modify these behaviors which prevent them from participating in primary care [15].

The main objective of this study was to determine whether profiles of adolescents could be identified based on their sociodemographic and behavioral characteristics and their PAS use. The secondary objective was to identify the obstacles and expectations of adolescents regarding the identification and management of their PAS use by their GP according to the profiles established.

## 2. Methods

### 2.1. Type of Study

This was a cross-sectional and descriptive study among a population of adolescents aged 12 to 17 years who were followed up in general practice in the Puy-de-Dôme French department.

### 2.2. Population Concerned

Adolescents were recruited from general practices in the Puy-de-Dôme French department.

The inclusion criteria were to be an adolescent aged between 12 and 17 years, followed by a GP volunteering to participate in the study, regardless of the initial reason for consultation.

The noninclusion criteria were adolescents unable to complete the questionnaire on their own or with difficulties in understanding the French language and adolescents who refused to participate in the research.

### 2.3. Method of Investigation

A random selection of participating GP practices was made from the list of private GPs in the Puy-de-Dôme, source National Institute of Statistics and Economic Studies (INSEE), 2016. The GPs were contacted by telephone in the order of the draw to explain the objectives of the study. If they refused, the next GP on the list was contacted. If a GP was drawn from a group practice, the agreement of all GPs in the practice was sought.

The questionnaires were freely available, either in the waiting room, in the secretary’s office, or directly in the GP’s office, depending on the layout of the practice (Appendix A). The GP or secretary could inform adolescents that a questionnaire was available as part of a study but should not encourage them to participate. A completed questionnaire attested to the adolescent’s consent. An opaque box was provided in each practice for the deposition of completed questionnaires to ensure the anonymity of the adolescents.

The questionnaire was presented as follows:-Number of times adolescents had visited their GP in the year.-Consumption of different PASs by adolescents and their consequences according to the screening scale for problematic alcohol and drug use among adolescents (DEP-ADO) [16].-An ad hoc multiple-choice questionnaire exploring adolescents’ expectations of their relationship with their GP, particularly regarding PAS, the various possible contacts for PAS consumption, their point of view on the use of a questionnaire as a tool for identifying PAS consumption, their point of view on the place of parents during a GP consultation, and, last, their knowledge of associations providing assistance to young people and adolescents.-Sociodemographic data and the adolescents’ lifestyle: gender, age, level of education, and urban or rural habitat.-This study was approved by the ethics commission of the academic department of general medicine of the Midi-Pyrénées region, France, under number 2018-009.

### 2.4. Screening Scale for Problematic Alcohol and Drug Use among Adolescents: DEP-ADO

The DEP-ADO is recognized as a reliable and valid tool for screening young people for problematic or risky alcohol and drug use [16].

The questions in the DEP-ADO cover alcohol and drug use in a 12-month period and in a 30-day period, regular alcohol and drug use (use at least once a week for at least one month), age of onset of regular alcohol or drug use, binge drinking, intravenous use of PAS, and the impact of use on various areas of the adolescent’s life (social relationships, delinquency, etc.) in a 12-month period.

A score is calculated and interpreted as follows: 0 to 13 points = adolescents with no obvious problematic use; 14 to 19 points = adolescents with emerging problems for whom front-line intervention is recommended; ≥20 points = adolescents who present significant substance use problems and for whom a specialized intervention is necessary.

### 2.5. Statistical Analysis

Statistical analysis was performed using Stata (version 15; StataCorp, College Station, TX, USA) and R 3.5.1 (http://cran.r-project.org/, accessed on 23 February 2021) software. All tests were two-sided, with an alpha level set at 0.05. Categorical data were expressed as the number of adolescents and associated percentages, and continuous data were expressed as the mean ± standard deviation or median [25th; 75th percentiles]. The consumption of each PAS was described as a percentage with its 95% confidence interval (CI). Comparisons of two independent groups (e.g., users versus nonusers, female versus male) were carried out by the chi-squared test or Fisher’s exact test for categorical variables and Student’s *t*-test or Mann–Whitney test for quantitative variables. Comparisons of three independent groups (e.g., 1 versus 2 versus 3 PAS consumed) were carried out by the chi-squared test or Fisher’s exact test for categorical variables and ANOVA or Kruskal–Wallis test for quantitative variables. A multiple correspondence analysis (MCA) followed by a mixed unsupervised classification (k-means clustering applied to the partition obtained from an ascending hierarchical classification using Ward’s distance) was also implemented to (i) study the relations between the modalities of the variables and (ii) determine the profiles of adolescents (clusters of individuals sharing closely similar characteristics). For this analysis, the variables were chosen according to univariate results, to clinical relevance, and to statistical distribution (characteristics always present or always absent were not considered). Only adolescents without missing data were included in the MCA, and the sample of excluded adolescents was compared to the sample of included adolescents as previously described.

## 3. Results

From March 2018 to April 2018, 60 GP practices were selected at random; 35 GP practices were in favor of participating in the study, and 28 GP practices included adolescents. They included 277 adolescents (mean age 14.5 ± 1.7 years), of which 155 (56%) were girls. There were 113 individuals (41%, 95% CI: 35 to 47%) who had used a PAS at least once in the past 12 months (Table 1).

The characteristics of the population are described in Table 1. There were no differences between users and nonusers except for age and the level of education.

### 3.1. Analysis of Consumption by Gender and PAS Consumed

Alcohol was the most commonly used PAS, followed by tobacco and cannabis, regardless of gender. The proportion of adolescents reporting having used one of these three substances at least once in the last 12 months was higher among boys than among girls. The data are presented in Figure 1.

### 3.2. Users’ Analysis

The description of adolescent users is presented in Table 2. Among the 113 adolescent users, the mean age of first use of all PAS was 13.7 ± 1.8 years, without any difference between girls and boys. Regarding the mean age of first use, only cannabis showed a significant difference according to gender (13.4 ± 1.4 years for girls and 14.9 ± 1.2 years for boys, *p* = 0.005). Recent PAS use (use in the last 30 days) concerned 43% (49/113) of adolescent users. Among the adolescent users, 8 (7%) had a DEP-ADO score ≥ 20, suggesting significant substance use problems and for whom a specialist intervention was necessary.

Among the 113 PAS users, 68 (60%) used only one product among tobacco, alcohol, and cannabis, 20 (18%) used two out of three, and 25 (22%) used all three. Among the 68 adolescents who used only one substance, 63 used only alcohol, 4 used only tobacco, and 1 used only cannabis.

Adolescents who used these three PAS had consumed earlier than the other users, had experimented with other drugs, had more heavy binge drinking, and had a higher DEP-ADO score, indicating a risk of substance abuse or misuse of PAS (Table 3).

### 3.3. Multiple Correspondence Analysis (MCA)

The MCA was based on the 243 subjects who had no missing data for the variables included in the analyses, and 43 were removed. These two samples were similar in age, gender, level of education, and level of PAS use.

Three groups were thus identified, whose characteristics are shown in Table 4: the group of nonusers (n = 134); the group of intermediate users (mainly alcohol) (n = 71); and the group of users at risk of substance abuse or misusing (n = 38).

The group of nonusers was mainly composed of girls aged between 12 and 14 years, and 59% were willing to open a dialog with their parents about the use of PAS. The group of intermediate users was made up mainly of girls aged 15 to 17 who consumed alcohol mainly and almost exclusively without binge drinking, and 65% were willing to discuss their PAS use with their parents. Finally, in the group of users at risk of substance abuse or misusing, the majority were boys aged 17 who were not in general education middle or high school. All these users were polydrug users, and 68% were binge drinkers. The age of first use was less than 14 years for 79% of this group. The DEP-ADO score was ≤13 for 61%. This was the only group that was mostly opposed (63%) to dialog with their parents.

### 3.4. Analysis of the Relationship with Their GP according to the Adolescents’ Use Profiles

The at-risk group of adolescents had consulted their GP more often than the others (3 [2; 4] consultations in the last 12 months in the group of nonusers, 3 [2; 5] in the group of intermediate users, and 4 [3; 5] in the at-risk group of misuse, *p* = 0.03), had been asked about their PAS use by their GP more often, and had also talked about their PAS use with their GP more often. The data are presented in Figure 2.

Regardless of which group they belonged to, adolescents felt that their GPs were attentive and responsive to their PAS use. They considered their GP to be in the right role and thus competent enough to ask adolescents about their use of PAS, while taking the time to ask the appropriate questions. They were not afraid of disappointing their doctor and, on the contrary, felt that their doctor understood them.

However, the at-risk adolescents were less confident and less comfortable, felt more judged, and were more afraid of the GP telling their parents. Despite this, adolescents in the at-risk group were the most willing to talk to their GP about their use of PAS, and almost half of the adolescents surveyed found it useful to use a questionnaire to discuss these topics, regardless of the group studied. The data are presented in Figure 3.

## 4. Discussion

Our study provides a description of 277 adolescents aged 12 to 17 years who were followed up in general practice. Among them, 113 (41%) reported having used a PAS at least once in the past 12 months. Alcohol was the most commonly used substance, followed by tobacco and cannabis, regardless of gender. Polydrug use of alcohol, tobacco, and cannabis was found among 25 users (22%). These users had used earlier than other users, had a higher risk of using other substances, used in the past 30 days, binge drank more, and had a higher DEP-ADO score than the rest of the sample.

Three profile users were identified. The group of nonusers consisted mainly of girls aged between 12 and 14 years, and 59% were willing to open a dialog about PAS use with their parents. The group of intermediate users was mainly composed of girls aged 15 to 17 who mainly drank alcohol without binge drinking, and 65% were willing to discuss their PAS use with their parents. The group of users at risk of substance abuse or misusing alcohol comprised mainly 17-year-old boys who were not in general education. All these users were polydrug users, and 68% were binge drinkers. The age of first use was <14 years for 79% of this group. The DEP-ADO score was ≤13 for 61%. This was the only group that was mostly opposed (63%) to dialog with their parents, and they consulted their GP more than the other two groups.

Regardless of the group to which they belonged, adolescents felt that their GPs were attentive, competent, and responsive to their PAS use. They considered their GP to be in the right role and taking the time to ask appropriate questions. The at-risk adolescents were less confident and less comfortable, felt more judged, and were more afraid of the GP telling their parents (in France, the civil majority is fixed at 18 years). Despite this, adolescents in the at-risk group were the most willing to talk to their GP about their PAS use. Almost half of the adolescents surveyed found it helpful to use a questionnaire to discuss these topics, regardless of the group studied.

Adolescence is the main period for experimenting with alcohol, tobacco, and cannabis [17]. Alcohol remains the first substance to be used during adolescence, followed by tobacco. Cannabis sees its experimentation and use develop and increase mainly during the “high school years”. It is still the most common illicit product, and experimentation with other illicit substances remains more confidential [17]. These findings were also observed in our sample.

To the best of our knowledge, no previous study has examined the use of PAS by adolescents in general practice. The reference studies are also conducted on a specific age and not on several age groups, which makes comparisons difficult. We know that between the ages of 11 and 13, experimentation is more frequent among boys and that this is no longer the case at age 15, where the difference between girls and boys remains only for cannabis experimentation [18]. In our study, the frequency of use of the various PAS appears to be lower than in the reference studies, both for first-time experimentation and for regular, daily use [1,4,19]. The difference in use between girls and boys also seems more marked in our sample. The age of first experimentation for tobacco and cannabis is lower than in the reference studies [1]. This is not the case for alcohol [3].

According to the 2009 GP Health Barometer, 63.2% of GPs state that they discuss the issue of tobacco use at least once with each patient, compared to 23% for alcohol and 7.8% for cannabis. Doctors declare that they discuss the latter two topics more often with patients they consider to be at risk: 72.7% of doctors for alcohol and 66.5% for cannabis [20]. More than 98% of GPs state that prevention is part of their role in the fields of smoking and alcoholism, and 92% of GPs said that prevention was part of their role in the field of cannabis [12]. In our study, the at-risk group of adolescents was interviewed and talked about their PAS use with their GP more often, which seems to suggest that GPs target their screening to adolescents who they judge to be at risk of substance abuse or misusing.

Given the specific vulnerability to the occurrence of misuse or its consequences, adolescents should be subject to particular surveillance with regard to the level of consumption of PAS [7,8]. However, in our study, we found a lack of screening. Despite the low level of communication between adolescents and GPs about PAS use, it is the adolescents who use the most (group 3) who have talked about it the most and/or who are the most willing to talk about it. Being empathetic and reminding patients of the principles of trust and confidentiality at each consultation would make it easier for GPs to overcome adolescents’ resistance to communication about their use of PAS. One other solution could be the use of questionnaires during the consultation, as almost half of the adolescents interviewed would have found it helpful to use a questionnaire to discuss these topics. For example, it should be noted that the subject of cannabis is declared easier to tackle by doctors who use questionnaires on this subject [12]. Thus, 41.2% of GPs who use preestablished questionnaires on cannabis find it very easy to discuss it with patients, compared with 19.1% of those who do not (*p* < 0.05) [12]. More generally, in 2009, 62.3% of doctors stated that they used preestablished questionnaires during their consultations to help identify the problem. These questionnaires mainly concern tobacco (34% of all GPs), alcohol (12.9%), and cannabis and other drugs (2.4%) [12]. It should be noted that this practice appears to have increased significantly since 1998 [20].

Our study therefore provides important information on the consumption of PAS by adolescents followed up in general practice and opens the way to early identification of the risks of substance abuse. The strength of this study lies in its original approach, with a study carried out directly in general practices, with adolescents consulting their GP, regardless of the initial reason for consultation. This approach presents the interest of an unprejudiced, observation-based approach. However, this technique has certain limitations that must be considered when interpreting the results. Indeed, it seems difficult to consider this sample as representative of the population studied, notably because of a possible recruitment bias due to the need for the GPs’ agreement to carry out the study in their practices. The desired minimum number of participants of 600 was not reached. Finally, social desirability bias, responses by bravado, and refusal to respond may lead to incorrect classification, thus limiting the scope of the data in this survey.

With their holistic view of the patient, GPs are therefore in the best position to identify adolescents in fragile situations and to help parents without making them feel guilty or trivializing them. However, many GPs, who are in the best position to identify risky use of PAS by adolescents at an early stage, do not feel competent or feel isolated in this field [6]. These topics are therefore rarely discussed. However, the treatment of PAS users appears to be particularly linked to the doctor’s propensity to address this issue without waiting for a request from the patient [20]. The questionnaire is therefore likely to be an aid for the practitioner less at ease with addictive practices. The short scales seem less tiring because they are quick to administer, and the standardized tools make it possible to discuss the question of PAS use with the patient independent of any presumption of misuse, insofar as they are supposed to be offered to everyone [21,22].

## 5. Implications and Contribution

Regardless of group, adolescents felt that their general practitioner was attentive, competent, understanding, and took the time to ask the appropriate questions. The at-risk group was the most willing to talk to their general practitioner. Almost half of the adolescents surveyed found it useful to use a questionnaire to discuss these topics.

## 6. Conclusions

In our study, 113 adolescents had used a PAS in the past 12 months. Alcohol was the most commonly used PAS, followed by tobacco and cannabis. Three groups were identified. Regardless of the group, adolescents felt that their GPs were attentive, responsive, competent, and understanding, and took the time to ask the appropriate questions in their roles. The at-risk group was less confident and less comfortable, they felt more judged, and were more afraid of the GP telling their parents. Despite this, the at-risk group was the most willing to talk to their GP. Almost half of the adolescents surveyed found it useful to use a questionnaire to discuss these topics.

This exploration of the professional practices of GPs shows the gradual recognition of the need to set up screening for substance abuse as early as possible and the use of tools for identifying addictive behavior in general practice.

## Figures and Tables

**Figure 1 ijerph-19-13231-f001:**
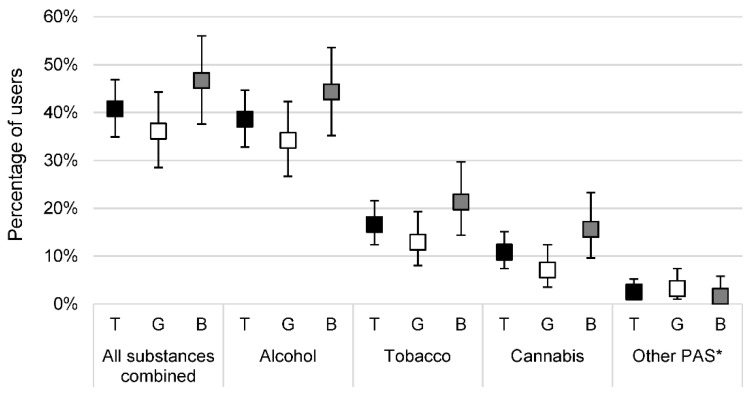
Distribution of use in the last 12 months according to psychoactive substances used and gender among the 277 adolescents. Data are presented as percentages and 95% confidence intervals. B: boys; G: girls; PAS: psychoactive substances; T: total. * At least one use of psychoactive substances among cocaine, glue, lysergic acid diethylamide (LSD), heroin, and/or amphetamines.

**Figure 2 ijerph-19-13231-f002:**
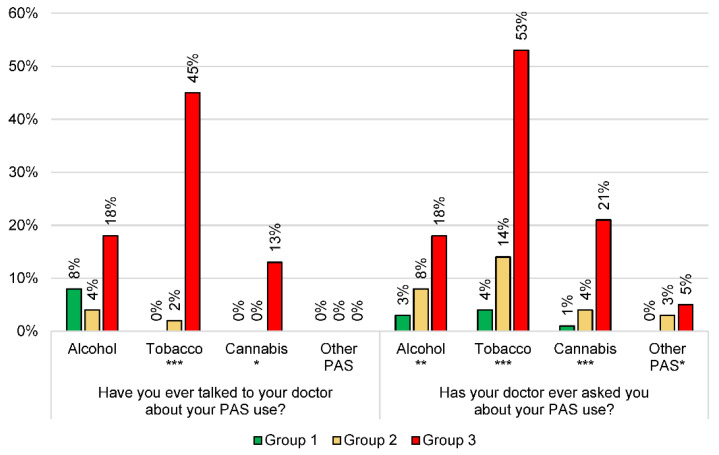
Analysis of the 243 subjects’ discussions with their GPs about their PAS consumption and according to their consumption profile. * *p* < 0.05; ** *p* < 0.01; *** *p* < 0.001. PAS: psychoactive substances. Other PAS: at least one use of psychoactive substances among cocaine, glue, lysergic acid diethylamide (LSD), heroin, and/or amphetamines.

**Figure 3 ijerph-19-13231-f003:**
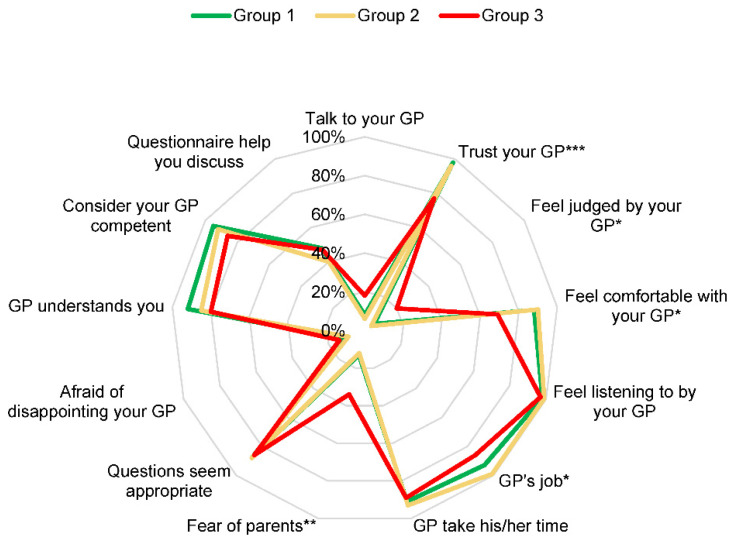
Factor analysis of the 243 subjects according to their relationship with their GP and according to their consumption profile. * *p* < 0.05; ** *p* < 0.01; *** *p* < 0.001. Talk to your GP: “Would you like to talk to your general practitioner about your psychoactive substances use?”; Trust your GP: “Do you trust your general practitioner?”; Feel judged by your GP: “Do you feel judged by your general practitioner?”; Feel comfortable with your GP: “Do you feel comfortable with your general practitioner?”; Feel listening to by your GP: “Do you feel listened to by your general practitioner?”; GP’s job: “Is it your general practitioner’s job to be interested in your psychoactive substances use?”; GP take his or her time: “Does your general practitioner take his or her time when looking into your psychoactive substances use?”; Fear of parents: “Are you afraid your general practitioner will tell your parents about your psychoactive substances use?”; Questions seem appropriate: “Do the questions from your general practitioner seem appropriate?”; Afraid of disappointing your GP: “Are you afraid of disappointing your general practitioner?”; GP understands you: “Do you feel that your general practitioner understands you?”; Consider your GP competent: “Do you consider your general practitioner competent?”; Questionnaire help you discuss: “Would a questionnaire, to be filled out during a consultation, help you discuss these topics with your doctor?”.

**Table 1 ijerph-19-13231-t001:** Description of the sample according to user profile, all substances combined.

	Total (N = 277)	Users * (N = 113)	Nonusers ^†^ (N = 164)	*p* Value
Female gender	155 (56%)	56 (50%)	99 (60%)	0.08
Age (years)	14.5 ± 1.7	15.6 ± 1.4	13.8 ± 1.4	**<0.001**
Housing with parents or family	273 (99%)	110 (97%)	163 (99%)	0.31
Urban localization	260 (94%)	104 (92%)	156 (95%)	0.29
Level of education				**<0.001**
Middle school	161 (58%)	32 (28%)	129 (79%)	
General high school	57 (21%)	37 (33%)	20 (12%)	
Vocational school	32 (12%)	22 (19%)	10 (6%)	
Technical school	18 (6%)	13 (12%)	5 (3%)	
Apprentice training center	5 (2%)	5 (4%)	0 (0%)	
Other ^‡^	4 (1%)	4 (4%)	0 (0%)	

Data are presented as number of adolescents (associated percentages) or mean ± standard deviation. * Use of at least one psychoactive substance in the last 12 months. ^†^ No use of psychoactive substances in the last 12 months. ^‡^ Adapted initial training course, job search, pregnancy, college. Significance of bold values: *p*-value indicating a statistically significant difference. The level of significance is defined at 5% (or 0.05).

**Table 2 ijerph-19-13231-t002:** Comparison of adolescent substance use behavior by gender.

	Total (N = 113)	Girls (N = 56)	Boys (N = 57)	*p* Value
Age of first use of any PAS (years)	13.7 ± 1.8	13.8 ± 1.9	13.6 ± 1.8	0.44
Age of regular use of any PAS (years) (n = 35)	14.1 ± 2.0	13.4 ± 2.5	14.6 ± 1.4	0.20
PAS use in the last 30 days	49 (43%)	22 (39%)	27 (47%)	0.39
Number of alcohol use of ≥5 drinks on the same occasion during the last 12 months	0 [0; 3]	0 [0; 1]	1 [0; 6]	0.02
DEP-ADO score				0.21
≤13	95/112 (85%)	49 (87%)	46/56 (82%)	
14–19	9/112 (8%)	2 (4%)	7/56 (13%)	
≥20	8/112 (7%)	5 (9%)	3/56 (5%)	

Data are presented as number of adolescents (associated percentages), mean ± standard deviation or median [25th; 75th percentiles]. DEP-ADO: screening scale for problematic alcohol and drug use among adolescents; PAS: psychoactive substances.

**Table 3 ijerph-19-13231-t003:** Comparison of the 113 users of psychoactive substances according to the number of substances used.

	1 PAS * (N = 68)	2 PAS * (N = 20)	3 PAS * (N = 25)	*p* Value
Female gender	38 (56%)	8 (40%)	10 (40%)	0.26
Age of first use of any PAS (years)	14.0 ± 1.8	13.7 ± 1.6	12.9 ± 1.9	**0.03**
Age of regular use of any PAS (years) (n = 35)	15.5 ± 0.7	14.3 ± 2.6	13.8 ± 1.6	0.16
Use of another PAS ^†^	0 (0%)	0 (0%)	7 (28%)	**<0.001**
PAS use in the last 30 days	19 (28%)	9 (45%)	21 (84%)	**<0.001**
Number of binge drinking ^‡^ during the last 12 months	0 [0; 1]	1 [0; 1]	6 [3; 18]	**<0.001**
DEP-ADO score				**<0.001**
≤13	66/67 (99%)	20 (100%)	9 (36%)	
14–19	1/67 (1%)	0 (0%)	8 (32%)	
≥20	0/67 (0%)	0 (0%)	8 (32%)	

Data are presented as number of adolescents (associated percentages), mean ± standard deviation or median [25th; 75th percentiles]. DEP-ADO: screening scale for problematic alcohol and drug use among adolescents; PAS: psychoactive substances. * Among alcohol, tobacco, and cannabis. ^†^ At least one use of psychoactive substances among cocaine, glue, lysergic acid diethylamide (LSD), heroin, and/or amphetamines. ^‡^ Consumption of at least five units of alcohol on the same occasion. Significance of bold values: *p*-value indicating a statistically significant difference. The level of significance is defined at 5% (or 0.05).

**Table 4 ijerph-19-13231-t004:** Multiple correspondence analysis of the 243 subjects according to their profile of psychoactive substance use.

	Group 1	Group 2	Group 3	*p* Value
	(n = 134)	(n = 71)	(n = 38)	
Female gender	77 (57%)	46 (65%)	13 (34%)	**0.008**
Age (years)				**<0.001**
12	33 (25%)	0 (0%)	0 (0%)	
13	45 (34%)	1 (1%)	1 (3%)	
14	37 (28%)	1 (1%)	8 (21%)	
15	18 (13%)	19 (27%)	5 (13%)	
16	0 (0%)	31 (44%)	2 (5%)	
17	1 (1%)	19 (27%)	22 (58%)	
Level of education				**<0.001**
Middle school	126 (94%)	6 (8%)	10 (26%)	
General high school	3 (2%)	38 (54%)	8 (21%)	
Other *	5 (4%)	27 (38%)	20 (53%)	
Alcohol use in the last 12 months	9 (7%)	51 (72%)	37 (97%)	**<0.001**
Tobacco use in the last 12 months	3 (2%)	4 (6%)	35 (92%)	**<0.001**
Cannabis use in the last 12 months	0 (0%)	1 (1%)	26 (68%)	**<0.001**
Other PAS use ^†^ in the last 12 months	0 (0%)	0 (0%)	6 (16%)	**<0.001**
Polydrug use of PAS in the last 12 months	0 (0%)	3 (4%)	38 (100%)	**<0.001**
Age of first use of any PAS (years)				**<0.001**
≤14	12 (9%)	18 (25%)	30 (79%)	
≥15	0 (0%)	35 (49%)	8 (21%)	
Not concerned	122 (91%)	18 (25%)	0 (0%)	
Binge drinking ^‡^	0 (0%)	17 (24%)	26 (68%)	**<0.001**
DEP-ADO score ≥ 14	0 (0%)	1 (1%)	15 (39%)	**<0.001**
Talk to their parents about their PAS use	79 (59%)	46 (65%)	14 (37%)	**0.02**
Talk to another family member about their PAS use	45 (34%)	35 (49%)	8 (21%)	**0.009**

Data are presented as number of adolescents (associated percentages). DEP-ADO: screening scale for problematic alcohol and drug use among adolescents; PAS: psychoactive substances. * Vocational school, technical school, apprentice training center, adapted initial training course, job search, pregnancy, college. ^†^ At least one use of psychoactive substances among cocaine, glue, lysergic acid diethylamide (LSD), heroin, and/or amphetamines. ^‡^ Consumption of at least five units of alcohol on the same occasion. Significance of bold values: *p*-value indicating a statistically significant difference. The level of significance is defined at 5% (or 0.05).

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
