# Peer review of "Barriers and Expectations of Adolescents Regarding the Identification and Management of Their Psychoactive Substance Use by Their General Practitioner"

_ijerph, 2022, doi:10.3390/ijerph192013231_

Round 1

Reviewer 1 Report

Thank you.

Best regards.

Author Response

  1. It appears odd to use a number before "Purpose", "Methods" etc... I invite also Authors to use the term "Aim" or "Aims" instead of purpose. Also, if allowed by the number of words for the Abstract, I invite Authors to draft a "Background" line (breadth of the problem, risk of short-, medium- and long-term consequences on physical and mental health). The crucial role of the primary care physician, especially with adolescents, should also appear somewhere in the abstract.

Response

We thank the reviewer for this remark which has allowed us to improve the abstract.

In the version submitted online, we did not include a number before "Purpose", "Methods", etc... It must be an automatic addition of the online submission software. We agree with you that this presentation may appear odd and we have removed the numbers.

We have followed your suggestion to change the term "Purpose" to "Aims".

We thank the reviewer for this very constructive and helpful comment regarding "Context". However, the abstract should normally be around 200 words and we are already exceeding this recommendation. We still decided to add a sentence about the crucial role of the primary care physician, which we also consider essential.

  1. I suggest replacing "Addictive" with "Addiction" and "Behavior" with "Addictive behavior. These are terms used more frequently in online literature searches, which would facilitate the dissemination of the work and its possible use/citation by other Authors.

Response

We thank the reviewer for this remark. The revised manuscript has been modified accordingly.

  1. Line 52. The Authors rightly mentioned some mental health consequences concerning self-injurious gestures and suicidality. Also to enrich the theoretical background of the text, I urge the authors that these are not only consequences but also causes. Impulsivity and its endophenotypes, in young substance users, often linked to self-injurious gestures/suicidality as well as other risk behaviors such as binge-drinking mentioned later in the text, has its own neuro-biological reason: the imbalance between the maturation of the frontal lobes versus the amygdala. I propose that the Authors, in writing a very short paragraph, use the following recent paper (not limited to alcohol use), in which this neurobehavioral issue is reviewed in the introduction and discussion: Impulsivity and Impulsivity-Related Endophenotypes in Suicidal Patients with Substance Use Disorders: an Exploratory Study . International Journal of Mental Health and Addiction. 2021;19(5):1729-1744. doi: 10.1007/s11469-020-00259-3. It would give a sense of the focus of the in peer-review study: that is, of why the magnitude of the problem among adolescents, its causes and consequences (in addition to possible purely behavioral reasons such as desire for transgression, group imitation, etc...)

Response

We thank the reviewer for this remark which has allowed us to improve the introduction. We also thank the reviewer for the relevance of the proposed reference. After reading the article proposed by the reviewer and a quick review of the literature, we decided to add a short paragraph on these neurobiological concepts and their consequences.

However, our work does not concern this very complex theoretical background. We did not want to push the neurobiological explanations too far in this case to avoid giving a bad summary of this discussion on this subject.

“Given a specific vulnerability to the occurrence of misuse or its consequences, adolescents should be particularly monitored with respect to their level of PAS use (1,2). Addiction is often initiated during adolescence when the still incomplete decision-making control circuits have difficulty regulating emotional, impulsive, and conditioned responses (3,4). PAS misuse during this period can cause specific damage to the brain maturation processes and neuropsychological development of the adolescent (2,4). For all these reasons, adolescence is a period of particular vulnerability to the use of PAS, which requires regular vigilance and monitoring of their use and rapid intervention in case of misuse.”

  1. Windle M, Mun EY, Windle RC. Adolescent-to-young adulthood heavy drinking trajectories and their prospective predictors. J Stud Alcohol [Internet]. mai 2005 [cité 1 oct 2020];66(3):313‑22. Disponible sur: http://www.jsad.com/doi/10.15288/jsa.2005.66.313
  2. Welch KA, Carson A, Lawrie SM. Brain Structure in Adolescents and Young Adults with Alcohol Problems: Systematic Review of Imaging Studies. Alcohol Alcohol [Internet]. 1 juill 2013 [cité 1 oct 2020];48(4):433‑44. Disponible sur: https://academic.oup.com/alcalc/article/48/4/433/534074
  3. Quinn PD, Harden KP. Differential Changes in Impulsivity and Sensation Seeking and the Escalation of Substance Use from Adolescence to Early Adulthood. Dev Psychopathol [Internet]. févr 2013 [cité 22 sept 2020];25(1):223‑39. Disponible sur: https://www.ncbi.nlm.nih.gov/pmc/articles/PMC3967723/
  4. Costanza A, Rothen S, Achab S, Thorens G, Baertschi M, Weber K, et al. Impulsivity and Impulsivity-Related Endophenotypes in Suicidal Patients with Substance Use Disorders: an Exploratory Study. Int J Ment Health Addict [Internet]. 1 oct 2021 [cité 4 oct 2022];19(5):1729‑44. Disponible sur: https://doi.org/10.1007/s11469-020-00259-3

  1. Please, unless I made a mistake in not seeing it, clarify the questionnaire was either an ad hoc prepared questionnaire or already validated in other studies. Please also make explicit here whether there were only questions with "closed" answers (as it seemed to me from the figure legend) or whether there were also "open" questions.

Response

We thank the reviewer for this comment which has encouraged us to clarify this point. The questionnaire exploring the relationship between the adolescent and his or her general practitioner is an ad hoc prepared questionnaire composed of closed-ended multiple-choice questions.

Following your very pertinent comment, we have proposed our questionnaire as an appendix to the editor.

  1. Discussion and conclusion. Please, again unless I am mistaken in not having seen it, make explicit here whether the questionnaires that the Authors rightly hope will be used in screening (in addition to the fundamental doctor-patient relationship, of which the openended interview is the prerequisite and the most important part). are the same as those used for the peer-reviewed study or whether they could be others (and which ones).

Response

We thank the reviewer for this comment which has encouraged us to clarify this point. We discussed this point a great deal in our team.

Concerning the use of questionnaires to facilitate the screening of substance use in general practice, we cannot unfortunately cite any specific questionnaires. In fact, according to the literature, the use of questionnaires, whatever they may be, facilitates the detection of the use of psychoactive substances. Moreover, after rereading the articles cited, the questionnaires are not specified, and we cannot therefore formally answer this question.

It seems to us to be biased to quote the most common questionnaires (Fagerström, AUDIT, CAST...) in our discussion because this could lead to the belief that these are the only questionnaires that are useful in current practice. The subject here is not the validity or strength of a questionnaire but only the help they can provide in addressing the issue of screening in general practice.

  1. Rightly, the authors have pointed out the strengths of their study, which are real and interesting. I suggest, for methodological rigor, to list the limitations as well.

Response

We thank the reviewer for this remark, which has allowed us to specify this element.

The limitations of our study were specified just after the strengths. After discussion and rereading, we did not feel it was useful or relevant to say more about them. But we remain open to discussion if the reviewer considers that important limitations have been omitted.

Here is the paragraph that lists them.

“However, this technique has certain limitations that must be considered when interpreting the results. Indeed, it seems difficult to consider this sample as representative of the population studied, notably because of a possible recruitment bias due to the need for the GPs’ agreement to carry out the study in their practices. The desired minimum number of participants of 600 was not reached. Finally, social desirability bias, responses by bravado and refusal to respond may lead to incorrect classification, thus limiting the scope of the data in this survey.”

Reviewer 2 Report

It would be interesting to compare the results of the analyses with data from other countries in order to determine whether the high level of trust in the GP is due to the specificity in France, or is the general statement that young people prefer to talk to an independent person about alcohol and drug consumption. It would also be important to make this a standard part of all GP visits as soon as possible and to raise awareness among schools/school psychologists.

The article presented for review is an interesting report on the analysis barriers and expectations of adolescents regarding the identification and management of their psychoactive substance use by their general practitioner. The issues raised should be considered important. The results of the analysis carried out are shown in a clear and readable form. In my opinion, the text is well prepared. It would be interesting to compare the results of the analyses with data from other countries in order to determine whether the high level of trust in the GP is due to the specificity in France, or is the general statement that young people prefer to talk to an independent person about alcohol and drug consumption. It would also be important to make this a standard part of all GP visits as soon as possible and to raise awareness among schools/school psychologists.

Introduction

I propose to supplement the introduction of information from other countries on PAS consumption in the adolescents.

Methods

Can they declare the system in France for GP. Up to what age the children go to a paediatrician, where are the parents usually present, and from when is the GP?

Discussion

Authors have basically referred to the data from France, it would be interesting to also look at and compare the use of PAS among adolescents in other countries. Is there any data on how other countries have established communication with young people about alcohol and drugs?

Author Response

  1. I propose to supplement the introduction of information from other countries on PAS consumption in the adolescents.

Response

We thank the reviewer for this remark which has allowed us to improve the introduction.

The choice of quoting only French data for substance use by adolescents was voluntary. The aim was to be able to compare our sample and our results with a comparable population.

However, we understand the need to be able to compare the French data with an international value. A clarification has therefore been made in the Introduction to compare the French data with a European average.

“According to the 2015 ESPAC survey, regarding the situation of French adolescents compared to their European counterparts, their level of recent (in the last 30 days) tobacco use is higher than average: 26% vs. 22% (11th place out of 35 countries). The recent alcohol consumption of 16-year-olds in France is in line with the European average: 47% (15th place). Finally, at age 16, the French lead the European ranking for recent cannabis use (17%).”

  1. Can they declare the system in France for GP. Up to what age the children go to a pediatrician, where are the parents usually present, and from when is the GP?

Response

We thank the reviewer for this remark, which has allowed us to specify this element.

This is a very difficult question. In France, the follow-up of children since their birth can be done indifferently by a pediatrician or a general practitioner. This decision is left to the choice of the parents as well as the availability of health professionals in the parents ‘place of residence. The territorial network of general practitioners is more complete than that of pediatricians.

Generally, children followed by a pediatrician change for a general practitioner during their adolescence but there again there is no precise rule, and it is left to the appreciation of the parents.

However, whether it is with a pediatrician or a general practitioner, in most cases, adolescents are seen in the presence of their parents until they reach the age of 18 (age of legal majority).

  1. Authors have basically referred to the data from France, it would be interesting to also look at and compare the use of PAS among adolescents in other countries. Is there any data on how other countries have established communication with young people about alcohol and drugs?

Response

We thank the reviewer for this comment which has encouraged us to clarify this point.

The comparison of French data with European data was made in the introduction according to your comment. After discussion in the team, it did not seem relevant to repeat the same data in the discussion.

To our knowledge, there is no study that compares the level of trust adolescents have in their general practitioners regarding the use of psychoactive substances. In the present state of knowledge, we cannot therefore know whether these results are specific to France.
